# Whey Protein Isolate as a Substrate to Design *Calendula officinalis* Flower Extract Controlled-Release Materials

**DOI:** 10.3390/ijms25105325

**Published:** 2024-05-13

**Authors:** Natalia Stachowiak-Trojanowska, Weronika Walendziak, Timothy E. L. Douglas, Justyna Kozlowska

**Affiliations:** 1Faculty of Chemistry, Nicolaus Copernicus University in Torun, ul. Gagarina 7, 87-100 Torun, Poland; nat.sta@doktorant.umk.pl (N.S.-T.); weronika.pw@doktorant.umk.pl (W.W.); 2School of Engineering, Lancaster University, Lancaster LA1 4YW, UK; t.douglas@lancaster.ac.uk

**Keywords:** whey protein isolate, sodium alginate, gelatin, *Calendula officinalis* flower extract, microparticles, polymeric films

## Abstract

The use of natural active substances and the development of new formulations are promising directions in the cosmetic and pharmacy industries. The primary purpose of this research was the production of microparticles based on whey protein isolate (WPI) and calcium alginate (ALG) containing *Calendula officinalis* flower extract and their incorporation into films composed of gelatin, WPI, and glycerol. Both swollen and dry microparticles were studied by optical microscopy and their sizes were measured. Water absorption by the microparticles, their loading capacity, and the release profile of flower extract were also characterized. The films were analyzed by mechanical tests (Young’s modulus, tensile strength, elongation at break), swelling capacity, contact angle, and moisture content measurements. The presented data showed that the active ingredient was successfully enclosed in spherical microparticles and completely released after 75 min of incubation at 37 °C. The incorporation of the microparticles into polymer films caused a decrease in stiffness and tensile strength, simultaneously increasing the ductility of the samples. Moreover, the films containing microparticles displayed higher swelling ability and moisture content compared to those without them. Hence, the materials prepared in this study with *Calendula officinalis* flower extract encapsulated into polymeric microspheres can be a starting point for the development of new products intended for skin application; advantages include protection of the extract against external factors and a controlled release profile.

## 1. Introduction

Whey is a by-product of cheese manufacturing from bovine milk. Whey proteins are the main protein component of ruminant milk after caseins, and they constitute 20% of all proteins in milk. Whey protein occurs in three main forms: isolate (WPI), concentrate (WPC), and hydrolysate (WPH). These fractions differ in the percentages of proteins, lipids, and carbohydrates [1]. During the purification process, fat and lactose are removed from whey protein, yielding WPI, whose protein content is at least 90%. The main proteins consist of β-lactoglobulin, α-lactalbumin, glycomacropeptide, immunoglobulins, bovine serum albumin, lactoferrin, lysozyme, prosthetic peptones, and others [2]. However, their content varies depending on the season and type of produced cheese [3,4], composition and type of milk [5], and the nature of the WPI purification process (e.g., membrane-separation, filtration processes, ion-exchange chromatography) [6,7]. Exposure of whey proteins to elevated temperatures above 60 °C initiates structural changes in proteins, which lead to the formation of extensive hydrogel networks [8]. Irreversible heat-induced gelation results from peptide denaturation and aggregation processes through covalent intermolecular bonds and other intermolecular non-covalent interactions, such as hydrophobic and electrostatic interactions [9]. The pH of the solution and the ionic strength have a significant impact on the spatial structure of the protein and are thus of great importance during protein hydrogel formation [10]. WPI exhibits wide functionalities due to its emulsifying, gelling, foaming, and water-binding properties [11,12]. WPI is becoming an increasingly popular functional and active food ingredient because it is produced in very large amounts and demonstrates numerous health benefits to humans. WPI has been applied not only in the food industry, but also in the cosmetic and pharmaceutical industries and the preparation of biomaterials [13,14,15].

WPI, as a dairy industry by-product, constitutes a relatively cheap and versatile material for various uses, such as encapsulation and thin polymeric film preparation [16,17,18,19]. Microparticles are spherical particles intended to enclose various substances, such as extracts [20], drugs [21], vitamins [22], dyes [23], perfumes [24], etc., in a polymeric matrix, depending on their application. Different methods can be employed for the production of microparticles, such as emulsion [25], extrusion [26], coacervation [27], or spray drying [28,29]. The main determinants for selecting the proper production method and wall material are the morphology and physicochemical properties of the microparticles and the type of encapsulated substance [30]. The main advantages of encapsulation include the protection of enclosed substances from external factors and undesired reactions (e.g., oxidation or deactivation). Hence, encapsulation fulfills a dual function in that it simultaneously increases and maintains the stability of these substances. Further reasons for encapsulation are control and modification of the release rate of substances, separation of incompatible materials, as well as masking of organoleptic properties of substances such as color, taste, and odor [31,32].

To date, various research studies have been carried out to enhance the properties of thin polymer films by combining different polymers [33], adding plasticizers [34,35], or even microparticles [36,37]. However, to the best of our knowledge, there is no report on the incorporation of *Calendula officinalis* flower extract into microparticles made from WPI and the modification of films by the addition of such microparticles. *Calendula officinalis*, also known as pot marigold, is an annually flowering plant belonging to the Compositae family. Although it is native to the Mediterranean and the Middle East, it is grown in many countries and sometimes grows as a wild plant. The composition of its extract is complex; it mainly comprises carbohydrates, lipids, terpenoids, carotenoids, and phenolic compounds, including phenolic acids, tannins, coumarins, and flavonoids [38,39]. For this reason, *Calendula officinalis* preparations possess multiple activities, including antioxidant, antibacterial, antifungal, antiviral, anti-inflammatory, and wound healing activities [40,41].

The aim of the present study was the production of microparticles with *Calendula officinalis* flower extract and thin films using WPI and the investigation of their morphological and physicochemical properties. Microparticles were obtained from WPI and sodium alginate using an extrusion method and Ca^2+^ as a crosslinking agent; conversely, films were fabricated using gelatin, WPI, and glycerol, and further modified by calcium alginate microparticles (ALG). The ultimate goal is the development of new, highly effective materials intended for skin application. The isolation of the pot marigold flower extract in microspheres will enable its release in a controlled manner. These types of materials can form the basis for the design of new cosmetics (such as cosmetic masks) or new carrier systems for dermatological applications.

## 2. Results

### 2.1. Characterization of Microparticles

The appearance of dry and swollen microparticles based on WPI and ALG containing *Calendula officinalis* flower extract is shown in Figure 1. The morphological observations showed that the swollen microparticles were spherical in shape. They became less regular after drying. The swollen and dry samples possessed smooth surfaces. On the basis of optical microscope images, the appearance of samples appeared to be independent of their composition.

The prepared microparticles were characterized by water absorption and loading capacity of plant extract and measurement of their sizes (Table 1). Based on the presented data, the diameter of the WPI/ALG microparticles decreased by a factor of more than two after drying. The size of the swollen samples was approximately 2250 µm. Moreover, the analysis demonstrated that the obtained microparticles revealed a high water absorption capacity. 

The pot marigold extract release profile embedded in the microparticles based on WPI and calcium alginate in acetate buffer at 37 °C is shown in Figure 2. As one can see from Figure 2, the active substance encapsulated in the prepared microparticles was completely released after 75 min.

On the basis of the analyses of the prepared microparticles, samples consisting of 4% WPI and 0.5% calcium alginate were selected for inclusion in polymer films.

### 2.2. Materials Characterization

#### 2.2.1. Mechanical Properties

The values of Young’s modulus, tensile strength, and elongation at a break during the stretching of dry films and films soaked in PBS buffer (pH = 5.7) are shown in Table 2. The film thickness was measured before testing. The thickness of the films without microparticles was approximately 0.16 mm, whereas samples containing microparticles displayed a thickness of 0.21 mm. The measurements revealed that the mechanical properties differed due to changes in the films’ composition. Dry films composed of gelatin and glycerol had lower values of Young’s modulus (497 ± 78 MPa) and tensile strength (29 ± 2 N), as well as higher elongation at break (17 ± 2), which indicates that they were more flexible and broke later than the samples containing WPI (669 ± 83 MPa; 33 ± 4 N and 4 ± 1%, respectively). Incorporating microspheres into both GEL and GEL/WPI films led to a slight decrease in the values of Young’s modulus and tensile strength, while the values of elongation at break were slightly higher. Thus, the samples without the microspheres (GEL and GEL/WPI) were fractionally stiffer than those with the addition of microspheres (GEL + M(WPI 4% + ALG 0.5%) and GEL/WPI + M(WPI 4% + ALG 0.5%), respectively. Considering samples before soaking in PBS buffer, the highest Young’s modulus (669 ± 83 MPa) and tensile strength (33 ± 4 N) values were displayed by samples composed of gelatin, WPI, and glycerol, whereas the lowest values were displayed by the film containing gelatin, glycerol, and microparticles; in this case, Young’s modulus and tensile strength were 474 ± 62 MPa and 23 ± 3 N, respectively.

#### 2.2.2. Swelling Tests

Figure 3 shows the swelling percentage ratios of films prepared from gelatin, WPI, and glycerol with and without the addition of WPI microparticles, which were conducted during 3 h of incubation in PBS buffer (pH = 5.7).

The swelling took place at a constant rate. After 15 min, the protein-based films absorbed PBS buffer, increasing their weight by 260–280%. Three hours later, their weight increased up to 530% for the film based on gelatin and WPI and 580–590% for gelatin films and gelatin/WPI films containing microparticles.

#### 2.2.3. Contact Angle Results

The results of contact angle measurements for diiodomethane (D) and glycerol (G) for protein-based films are presented in Table 3. It was impossible to measure the contact angles of films containing microparticles due to their high surface roughness. A polymeric film composed of gelatin and glycerol displayed higher contact angles for both liquids, glycerol and diiodomethane (75.8 ± 0.4° and 52.8 ± 1.4°, respectively), than the film containing WPI, gelatin, and glycerol (71.4 ± 0.4° for glycerol and 50.4 ± 0.8° for diiodomethane). The addition of WPI to the film composition led to a change in non-covalent forces between the first monolayer of film and liquid and, therefore, decreased contact angles.

#### 2.2.4. Moisture Content

The results of moisture content after drying the samples in an oven at 110 °C to a constant weight are shown in Figure 4. Moisture content is a parameter connected with the volume occupied by water molecules in the microstructural network of the film.

According to the results, the film composition affects the samples’ moisture content. The highest moisture content was observed for the film composed of gelatin, glycerol, and microparticles (19%). In contrast, the lowest moisture content value was displayed by a sample containing gelatin, WPI, and glycerol (12.5%).

## 3. Discussion

The samples composed of 5% WPI and 0.5% ALG displayed the highest water absorption rate (approximately 830%). In turn, the lowest water absorption was displayed by the M(WPI 4% + ALG 1%) samples (approximately 670%). The loading capacity of *Calendula officinalis* flower extract into the variously formulated carriers was determined by the spectrophotometric method. As mentioned in other studies, the use of alginate alone leads to a low encapsulation efficiency [42,43]. This is due to diffusion through the porous structure of the hydrogels. However, a combination of the alginate with proteins improves the encapsulation of the active substance [43,44]. The results showed that the composition of the samples impacted the incorporation efficiency of the active ingredient. The largest amount of plant extract was entrapped in the M(WPI 5% + ALG 0.5%) microparticles (approximately 293 mg/g based on gallic acid). A lower content of calcium alginate in the microspheres is related to a higher loading capacity.

Phenolic compounds have been encapsulated in both polymeric micro- and nanoparticles in order to control their release rate in various media [45,46]. It is difficult to achieve the goal of targeted release. Therefore, the study of the behavior of the ALG/WPI microparticles containing pot marigold extract in an acidic environment is of great importance to gain a better understanding of its potential application in dermatology and cosmetics. The composition of the microspheres influenced the plant extract release rate. The microparticles composed of 0.5% calcium alginate showed a faster release rate of active ingredient than samples made from 1% of this polysaccharide. A two-stage release profile was observed for samples containing 1% calcium alginate. After 60 min, there was a rapid increase in release from these microparticles. In contrast, samples with 0.5% of polysaccharide exhibited a smooth release rate.

As expected, the soaking of materials led to a significant decrease in Young’s modulus and tensile strength values due to their hydration. The wet samples were significantly less stiff than the samples prior to soaking. The findings in the present study are consistent with other studies investigating the mechanical properties of protein-based films. The microstructural and physical properties of films composed of WPI and gelatin have been investigated [47]. It was noted that WPI exhibited a more twisted network microstructure (compared to the organized network of gelatin), which could improve the film’s mechanical strength and reduce its ductility. During the mixing of WPI and gelatin, the particles’ size may be reduced by the electrostatic attraction and hydrogen bonding (between the WPI amido and the gelatin carboxyl groups); hence, the formed chains could be thickened. Cao et al. evaluated the effect of soy protein isolate/gelatin ratio on the mechanical properties of composite films [48]. They also attributed the changes in tensile strength to the protein/protein intermolecular interactions determined by hydrogen bonds or by electrostatic interaction and/or by hydrophobic nature. The sequence of amino acid residues and the three-dimensional network influence these interactions. Pérez-Gago analyzed the influence of WPI denaturation time and temperature on the physical properties of WPI films plasticized with glycerol [49]. They found that with increasing heat-denaturation time (from 5 to 20 min) and temperature (from 70 to 100 °C), Young’s Modulus, tensile strength, and percentage elongation increased. This was attributed to the covalent disulfide bonding of the heat-denatured whey protein films created during the unfolding of globular whey protein, which resulted in stronger films that withstand greater deformations.

It can also be noted that the addition of microparticles caused a decrease in the values of Young’s modulus and tensile strength and a rise in elongation at break (for both dry and soaked films). This indicates that the addition of microparticles modified or disrupted the original structures of the polymeric matrix. The same observations have been reported in other papers [37,50]. Gelatin films modified with papaya peel microparticles showed lower Young’s modulus and tensile strength than the control sample due to lack of cohesion of residues with gelatin [36]. The authors of the latter study emphasized the importance of the cohesion of the polymer matrix constituents as the predominant reason for the film’s mechanical strength, causing a good interaction between the microparticles and the polymer matrix.

All prepared samples also contained glycerol, a plasticizer that reduces the intermolecular hydrogen bonding while increasing the intermolecular spacing and mobility of biopolymer chains [51]. It is assumed that the protein–protein interactions are being replaced by the polymer–plasticizer hydrogen bonds created by the plasticizer polar groups (−OH) [48]. Therefore, these interactions may be affected by the plasticizers’ molecular size, configuration, the total number of functional hydroxyl groups, and the selected polymer’s compatibility. Glycerol has been found to be one of the most effective plasticizers. Due to its small size, it can penetrate more easily between the polymer chains and weaken the interaction between polymer chains, thus increasing the material’s flexibility and extensibility [52].

The materials were not soluble in water, thus, it was possible to carry out the swelling measurements. The swelling degree is an indicator of the protein cross-linking degree. Swellability depends on the structure and properties of the solvent and the polymer, as well as the interactions between them [53]. It can be seen that the films composed of gelatin and glycerol displayed slightly higher swelling properties than the films with the addition of WPI. Moreover, the films containing microparticles also displayed higher water uptake. The insolubility of WPI may cause lower water uptake by protein-based films due to the intermolecular disulfide bonds formed during the heat-denaturation process [54]. Corresponding swelling ratios were observed in research performed by Amjadi et al. on WPI-based films containing nanoemulsions of orange peel essential oil for packaging purposes [55]. They observed a swelling ratio of ~1000% after 24 h of immersion in water. Esteghlal et al. investigated how the physical and mechanical properties of gelatin/carboxymethyl cellulose (CMC) films are affected by the electrostatic interactions between the biopolymers [56]. They found that the swelling properties are influenced by the mixing ratio and different pH values (swelling ratio ranged from 240 to 585%). Moreover, Cao et al. noticed that with increasing gelatin content in gelatin/soy protein isolate film, the degree of swelling capacity increased (from 400 to 950%) owing to the higher swelling properties of gelatin compared to the soy protein isolate (SPI) [48].

The surface free energies and their polar and dispersive components were determined using the Owens–Wendt method (Table 3). It can be seen that samples containing WPI had higher polar (6.8 mJ/m^2^) and dispersive (28.5 mJ/m^2^) components compared to the gelatin film (5.2 mJ/m^2^ for polar and 28.0 mJ/m^2^ for dispersive components). The surface free energies for gelatin and gelatin/WPI films were 33.2 mJ/m^2^ and 35.3 mJ/m^2^, respectively. Based on the low value of polar components, it is concluded that both films possessed less-hydrophilic surfaces; however, the film surface of the sample containing WPI displayed a slightly higher polarity. This can be ascribed to the intermolecular interactions between gelatin and WPI, which interfered with the orientation of polar groups toward the film surface. Glycerol, as well as hydroxyl, amino, and carboxyl groups between two polymers, participated in the formulation of hydrogen bonding. Furthermore, WPI and gelatin molecules could form compact aggregates through electrostatic interactions [57,58].

Films containing WPI showed a lower moisture content than the samples without WPI. However, the introduction of WPI/ALG microparticles into polymer films led to higher moisture content. Other researchers have also made similar observations; Shams et al. evaluated the moisture content of WPI/gelatin films modified by nanoclay and orange peel extract. The control film had a moisture content of approximately 34% [59]. The effect of glycerol, xylitol, and sorbitol on the physical properties of WPI films has also been investigated [60]. It was observed that samples plasticized with glycerol (from 40 to 60% depending on the plasticizer/protein ratio) displayed the highest moisture content, whereas the addition of xylitol and sorbitol resulted in a moisture content of 15–20%. WPI-based films have also been reported to display a moisture content of ~16.5% [54], whereas films containing gelatin displayed a moisture content of ~14.5% [61].

## 4. Materials and Methods

### 4.1. Materials

Whey protein isolate (WPI) (BiPRO, Davisco Foods International Inc., Eden Prairie, MN, USA) containing 97.7% protein, of which 75% was β-lactoglobulin by dry mass (according to the manufacturer’s specification) was used. Sodium alginate was supplied by BÜCHI Labortechnik AG (Flawil, Switzerland); the viscosity average molecular weight was determined in our laboratory as equal to 55,800 for K = 0.0178 cm^3^/g and a = 1 [62]. Gelatin type A (GEL) from porcine skin, Folin–Ciocalteu reagent, and gallic acid were acquired from Sigma-Aldrich (Poznan, Poland). The hydroglycolic *Calendula officinalis* flower extract (propylene glycol/water (80:20)) was obtained from Provital S.A. (Barcelona, Spain). All other reagents were obtained from Chempur (Piekary Śląskie, Poland). All used chemicals were of analytical grade.

### 4.2. Preparation of Microparticles

Microparticles (M) consisting of WPI and calcium alginate (ALG) were prepared using an encapsulator (B-395 Pro, BÜCHI Labortechnik AG, Flawil, Switzerland). Microparticles from solutions with different concentrations of WPI and sodium alginate were prepared. WPI solution with a concentration of 4% or 5% and sodium alginate solution with a concentration of 0.5% or 1% were used. First, a mixture of WPI and sodium alginate with the addition of 0.5% marigold flower extract was prepared. The ingredients were mixed on a magnetic stirrer for an hour at room temperature and then left without stirring for 2 h to ensure complete hydration of proteins. After this time, the polymer solutions with plant extract were heated for 40 min at 80 °C to denature the proteins contained in the WPI. The resulting solutions were cooled overnight at room temperature [63].

The production of microparticles using an encapsulator started by transferring the WPI and sodium alginate solution containing *Calendula officinalis* flower extract to a pressure bottle. Then, the mixture was forced through a 1000 µm diameter nozzle and separated into droplets by an electrical field. The formation of microparticles took place in a bath with a crosslinker solution (0.5 M CaCl_2_), which was continuously stirred to prevent the agglomeration of microparticles. The produced calcium alginate microspheres were kept in the bath with the crosslinking solution for 15 min. The collected microparticles were rinsed with distilled water and immersed in the extract.

Depending on the content of components, the obtained microspheres were named M(WPI 4% + ALG 0.5%), M(WPI 4% + ALG 1%), M(WPI 5% + ALG 0.5%), M(WPI 5% + ALG 1%).

### 4.3. Characterization of Microparticles

#### 4.3.1. Imaging of Microparticles

The appearance and sizes of the prepared microparticles were observed by optical microscope SMZ-171 BLED (Motic, Hong Kong, China) at a magnification of ×10. Imaging of swollen and dry polymer microspheres was performed. Drying of the samples lasted 72 h at room temperature. The images and diameters of the samples were recorded using Motic Images Plus 3.0 software.

#### 4.3.2. Water Absorption of Microparticles

Each type of the obtained microparticles was weighed after drying for 72 h and immersion in phosphate saline buffer (pH = 5.7) for 2 h. The test was performed in triplicate for all microparticle types. The water absorption capacity (1) was defined as the ratio of the increase in weight (swollen microparticles) (*W_w_*) to the initial weight (dry microparticles) (*W_d_*), as follows:(1)$$\mathrm{water}\;\mathrm{absorption}\;\left(\%\right)=\frac{\left(W_{w}-W_{d}\right)}{W_{d}}\times 100$$

#### 4.3.3. Loading Capacity of Microspheres

The loading capacity of the microspheres was determined by quantifying the phenolic compounds contained in the *Calendula officinalis* flower extract enclosed in the microspheres. For this purpose, the spectrophotometric method with Folin–Ciocalteu reagent was used [64]. The microspheres were weighed and immersed in 2 mL of 1 M NaOH for 1 h. After centrifuging the samples, the supernatant solution was collected. A total of 20 µL of sample with the extract was mixed with 1.58 mL of distilled water and 100 µL of Folin–Ciocalteu reagent was added. After 4 min, 300 µL of saturated Na_2_CO_3_ solution was added. The mixture was incubated for 30 min at 40 °C to obtain a typical blue color. The absorbance was measured at a wavelength of 725 nm using a UV-Vis spectrophotometer (UV-1800, Shimadzu, Kyoto, Japan). The presented results were calculated based on gallic acid using the standard curve equation. Three measurements were made for each type of sample.

#### 4.3.4. In Vitro Release

The release of extract entrapped in microparticles was also investigated by evaluation of phenolic content using a spectrophotometric method. Each type of microsphere was weighed and placed in acetate buffer (pH = 5.4). Samples were incubated at 37 °C. The solution was collected after 15, 30, 45, 60, and 75 min and the new portion of acetate buffer was added to the microspheres. Samples for measurement were prepared as in the previous Section 4.3.3, using the Folin–Ciocaltou reagent. Absorbance was measured at 725 nm with a UV-Vis spectrophotometer (UV-1800, Shimadzu, Kyoto, Japan) [65].

### 4.4. Preparation of Films with Microspheres

The films were fabricated from gelatin, WPI, and plasticizer (glycerol) using a solution casting technique [66]. The scheme of fabrication of gelatin/WPI/glycerol films with microparticles, as well as the preparation of WPI and calcium alginate microparticles, is presented in Figure 5. First, a solution of gelatin and WPI was prepared at concentrations of 4% and 2%, respectively, by mixing the ingredients on a magnetic stirrer for 1 h at room temperature. After this time, 2% (*w*/*v*) of glycerol was added and stirring was continued at 80 °C for 30 min. Then, a 5.5% suspension of the microspheres was added to the obtained solutions. After analyzing the prepared microparticles, the M(WPI 4% + ALG 0.5%) type was selected for location in films due to its smallest size in the swollen state. The mixtures were cast onto Petri dishes and allowed to dry at room temperature for 7 days. This matrix was denoted as GEL/WPI + M(WPI 4% + ALG 0.5%). The films were also prepared from a 4% gelatin solution following the procedure described above (GEL + M(WPI 4% + ALG 0.5%). For comparison, matrices without the addition of microparticles were prepared (GEL/WPI and GEL). The thickness of the obtained films was measured with a digital dial thickness gauge at a resolution of 0.001 mm (Sylvac, Yverdon-les-Bains, Switzerland).

### 4.5. Characterization of Films

#### 4.5.1. Mechanical Tests

Mechanical properties of the prepared films with and without microspheres were studied using a mechanical testing machine equipped with tensile grips (EZ-Test SX Texture Analyzer, Shimadzu, Kyoto, Japan). Specimens with initial dimensions of 50 mm in length and 4.5 mm in width were prepared by cutting with a dumbbell-shaped sharpener. The dry specimens and the specimens soaked for 5 min in PBS buffer (pH = 5.7) were examined. The prepared specimens were inserted between the machine clamps and stretched to break. The elastic modulus (Young’s modulus, E) was calculated from the slope of the stress–strain curve in the linear region. The tensile strength and the elongation at break of the films were also determined. The measurements were carried out at a velocity of 2 mm/min. The results were recorded using Trapezium X software (version 1.4.5, Shimadzu, Kyoto, Japan). Five measurements were made for each type of film.

#### 4.5.2. Evaluation of Swelling Capacity

The swelling ratio of the obtained films was tested by immersion in a phosphate saline buffer (PBS) at pH 5.7 for 3 h. The dry samples were weighed (*W*_1_) and placed in PBS solution. The measurements were conducted after 15 min, 30 min, 1 h, 2 h, and 3 h. After each interval, the samples were removed from the phosphate saline buffer and reweighed (*W*_2_) [67]. The swelling degree of the films was calculated using the following Equation (2):(2)$$\mathrm{swelling}\;\mathrm{degree}\;\left(\%\right)=\frac{\left(W_{2}-W_{1}\right)}{W_{1}}\times 100$$

#### 4.5.3. Contact Angle Measurements

The contact angles (°) of two liquids, diiodomethane (apolar liquid) and glycerol (polar liquid), on polymeric films were measured at constant room temperature (22 °C) using a DSA G10 goniometer equipped with a drop shape analysis system (Krüss GmbH, Wolfsburg, Germany). To obtain contact angle values, the average of five measurements was calculated. The surface free energy and its polar and dispersive components were calculated using the Owens–Wendt method [68].

#### 4.5.4. Moisture Content

The moisture content of the films with and without the addition of microparticles based on WPI and ALG was determined. Measurement of weight loss after drying in an oven at 110 °C was conducted to a constant weight [69]. After removal from the oven, the samples were stored in a desiccator. The samples were analyzed in triplicate. The moisture content (MC, %) was defined as the initial weight (*W_i_*) of each sample and the weight after drying (*W_d_*) using the Formula (3) below:(3)$$MC\;\left[\%\right]=\frac{W_{i}-W_{d}}{W_{d}}\times 100$$

#### 4.5.5. Statistical Analysis

One-way ANOVA with Tukey’s pairwise analysis was performed to statistically compare the results of microparticle (size, water absorption, and loading capacity) and film (mechanical properties and moisture content) characterization. GraphPad Prism 8 (GraphPad Software, San Diego, CA, USA) was used for all analyses. Data are shown as the mean ± S.D. for each experiment. *p*-values < 0.05 were considered significant. Statistically significant differences were marked with different superscript letters.

## 5. Conclusions

The focus of this study was to incorporate microparticles based on WPI and ALG containing *Calendula officinalis* flower extract into various WPI/gelatin-based films, mimicking a dermatological material for sustained, controlled delivery. Pot marigold extract was selected because of its beneficial antioxidant, anti-inflammatory, antimicrobial, and anti-viral properties. Microparticles consisting of 4% WPI and 0.5% ALG were incorporated into films. The WPI/gelatin-based films displayed enhanced mechanical strength, reduced ductility, slightly higher polarity, and lower moisture content compared to gelatin films. Furthermore, the microparticle-loaded samples demonstrated a higher capacity for water uptake and were less stiff than those without microparticles. The vital advantage of microparticles is the possibility to control the release rate of an active substance. The obtained results indicate the potential of GEL and GEL/WPI films modified with the addition of microspheres as material for cosmetic or dermatological applications. To confirm the functional properties and effectiveness of the films, measurements of skin parameters with the participation of volunteers are planned in the near future.

## Figures and Tables

**Figure 1 ijms-25-05325-f001:**
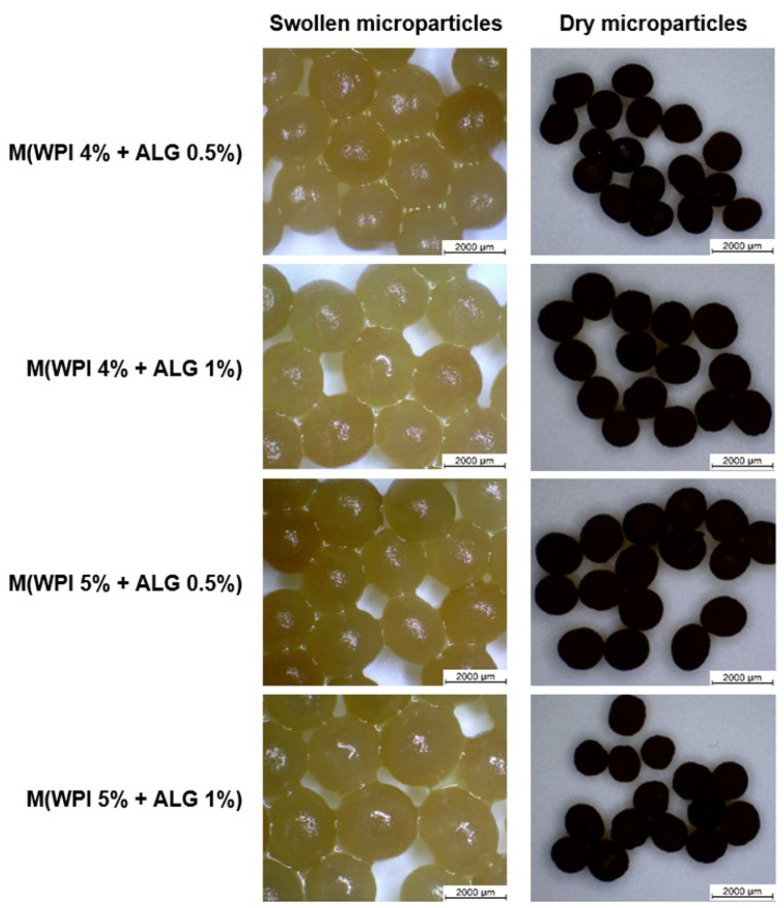
Microscope images of dry and swollen whey protein isolate (WPI) and calcium alginate (ALG) microparticles containing plant extract (scale bar 2000 µm).

**Figure 2 ijms-25-05325-f002:**
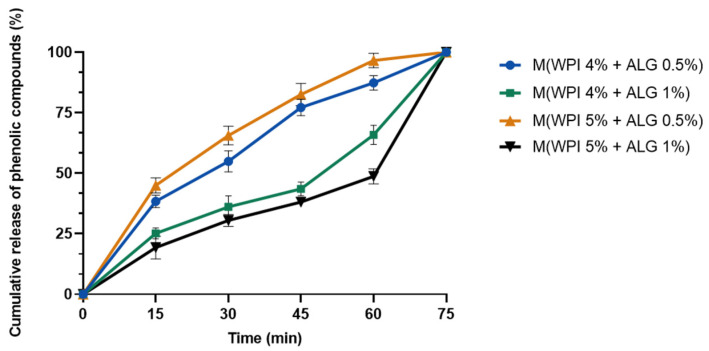
In vitro release of *Calendula officinalis* flower extract from microparticles (M) based on whey protein isolate (WPI) and calcium alginate (ALG).

**Figure 3 ijms-25-05325-f003:**
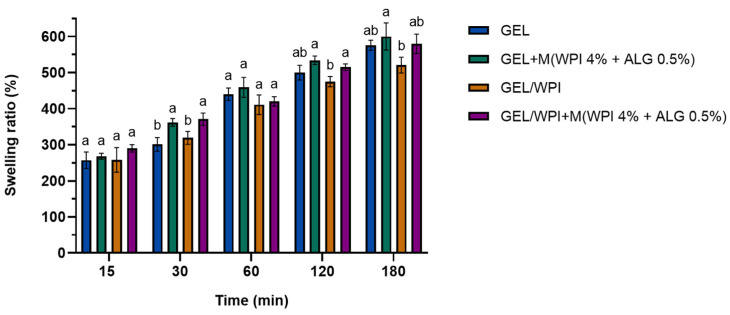
Swelling tests of the prepared films based on gelatin (GEL) and whey protein isolate (WPI) with and without microparticles (M) based on whey protein isolate (WPI) and calcium alginate (ALG). Different letters indicate a difference at *p* < 0.05. Therefore, the values labeled by one or more letters (a,b) indicate that variables in a column are statistically indistinguishable at *p* < 0.05 if they share at least one letter.

**Figure 4 ijms-25-05325-f004:**
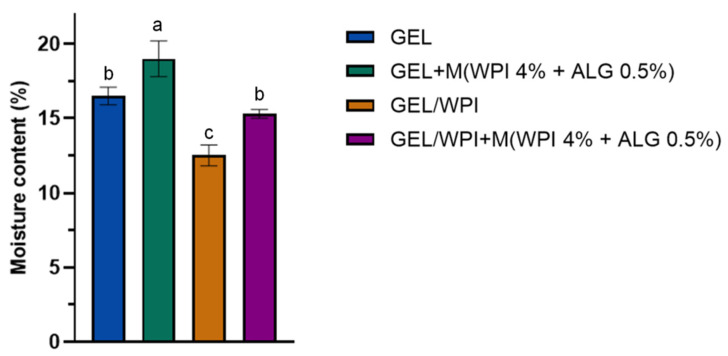
Moisture content (%) of the prepared polymer films based on gelatin (GEL) and whey protein isolate (WPI) with and without microparticles (M) based on whey protein isolate (WPI) and calcium alginate (ALG). Different letters indicate a difference at *p* < 0.05. Therefore, the values labeled by one or more letters (a–c) indicate that variables in a column are statistically indistinguishable at *p* < 0.05 if they share at least one letter.

**Figure 5 ijms-25-05325-f005:**
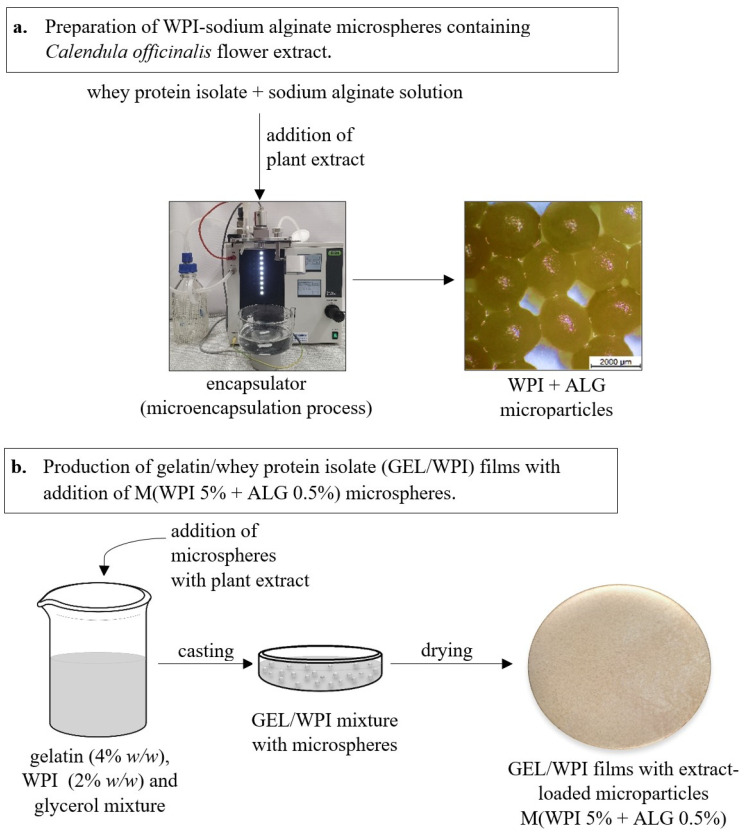
The preparation scheme of WPI and calcium alginate microparticles (**a**) and production of gelatin/WPI/glycerol films with microparticles (**b**).

**Table 1 ijms-25-05325-t001:** Characterization of the prepared microparticles: their sizes, swelling ratio, and loading capacity of *Calendula officinalis* flower extract. The values with different superscript letters in a column are significantly different (*p* < 0.05).

Microparticles	Particle Size (µm)		Water Absorption (%)	Loading Capacity (mg/g)
	Swollen	Dry		
M(WPI 4% + ALG 0.5%)	2183 ± 53 ^bc^	986 ± 32 ^a^	784 ± 45 ^a^	262 ± 7 ^b^
M(WPI 4% + ALG 1%)	2245 ± 43 ^ab^	993 ± 39 ^a^	669 ± 57 ^a^	169 ± 5 ^d^
M(WPI 5% + ALG 0.5%)	2339 ± 56 ^a^	982 ± 16 ^a^	829 ± 43 ^a^	293 ± 6 ^a^
M(WPI 5% + ALG 1%)	2201 ± 61 ^abc^	990 ± 30 ^a^	723 ± 38 ^a^	234 ± 9 ^c^

**Table 2 ijms-25-05325-t002:** Young’s modulus, tensile strength, and elongation at break of the dry and soaked polymer films with microparticles (WPI 4% + ALG 0.5%) and without them. Different superscript letters indicate a difference at *p* < 0.05.

Films	Young’s Modulus (MPa)		Tensile Strength (N)		Elongation at Break (%)	
	Dry	Soaked	Dry	Soaked	Dry	Soaked
GEL	497 ± 78 ^abc^	1.5 ± 0.21 ^a^	29 ± 2 ^abc^	0.34 ± 0.04 ^b^	17 ± 2 ^b^	63 ± 3 ^abc^
GEL + M(WPI 4% + ALG 0.5%)	474 ± 62 ^bc^	1.4 ± 0.52 ^a^	23 ± 3 ^bc^	0.33 ± 0.05 ^b^	21 ± 1 ^a^	70 ± 3 ^a^
GEL/WPI	669 ± 83 ^a^	1.1 ± 0.07 ^a^	33 ± 4 ^a^	0.54 ± 0.05 ^a^	4 ± 1 ^c^	57 ± 4 ^bc^
GEL/WPI + M(WPI 4% + ALG 0.5%)	518 ± 43 ^ab^	0.83 ± 0.05 ^a^	30 ± 3 ^ab^	0.42 ± 0.02 ^b^	7 ± 1 ^c^	68 ± 7 ^ab^

**Table 3 ijms-25-05325-t003:** The contact angles of diiodomethane (D) and glycerol (G), the surface free energy (γ_s_), polar (γ_s_^p^), and dispersive (γ_s_^d^) components for polymer films based on gelatin and whey protein isolate (calculated by Owens–Wendt method). Different superscript letters indicate a difference at *p* < 0.05.

Sample	Contact Angle (°)		Surface Free Energy (γ_s_) (mJ/m^2^)	Dispersive and Polar Components (mJ/m^2^)	
	G	D		γ_s_^d^	γ_s_^p^
GEL	75.8 ± 0.4 ^a^	52.8 ± 1.4 ^a^	33.2	28.0	5.2
GEL/WPI	71.4 ± 0.4 ^b^	50.4 ± 0.8 ^a^	35.3	28.5	6.8

## Data Availability

The data that support the findings of this study are available from the corresponding author upon request.

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
