# Peer review of "Whey Protein Isolate as a Substrate to Design Calendula officinalis Flower Extract Controlled-Release Materials"

_ijms, 2024, doi:10.3390/ijms25105325_

Round 1

Reviewer 1 Report

Comments and Suggestions for Authors

Comments and Suggestions for Authors:

In this paper, the authors produced microparticles with Calendula officinalis flower extract and thin films using whey protein isolate (WPI) and investigated their morphological and physico-chemical properties. The prepared new materials intended for application to the skin with high effectiveness. However, this paper has some problems that should be explained.

1.     Line 93, Section 2, Materials and Methods, containing 97.7% protein and 75% β-lactoglobulin in DM. The percentage of protein and lactoglobulin is greater than 100%? Please confirm.

2.     Line 110, 2.2. Preparation of Microparticles, stirring for 2h to ensure complete hydration of proteins. How to determine when complete hydration will occur in 2 hours?

3.     In 3.1. Characterization of Microparticles section. At the same ALG concentration, samples composed of 5% WPI seem to increase the water absorption rate. Why were only 4 and 5% chosen for the WPI ratio? Is there any reason?

4.     The description of Lines 233-235 was different from table 1:

“The samples composed of 5% WPI and 0.5% ALG displayed the highest swelling degree (approximately 830%)” à Does it refer to swelling degree or water absorption rate %?

“In turn, the lowest water absorption was displayed by M(WPI 5% + ALG 1%) samples (approximately 670%)” à M(WPI 4% + ALG 1%) = 669 ± 57%; M(WPI 5% + ALG 1%) = 723 ± 38%

5.     Line 264-265, “Summarizing the analyses of…samples consisting of 5% WPI and 0.5% calcium alginate were selected ...”, but the following studies used M(WPI 4% + ALG 0.5%). Is there anything wrong?

Table 2 annotation “…films with microparticles (WPI 5% + ALG 0.5%)”, but the table content was M(WPI 4% + ALG 0.5%).

6.     This manuscript does not test any skin applications, and the "potential skin application" in the title is not appropriate.

Comments on the Quality of English Language

 Moderate editing of English language required

Author Response

Thank you very much for your valuable feedback. You have provided valuable guidance on every part of the article. These opinions have greatly benefited our team. Your guidance has allowed us to re-examine the article. In terms of structure, writing ideas, logic, and other aspects, it has given us a new and profound understanding of the following work steps. Thank you again for your help to our team.

Reviewer 1

Comments and Suggestions for Authors:

In this paper, the authors produced microparticles with Calendula officinalis flower extract and thin films using whey protein isolate (WPI) and investigated their morphological and physico-chemical properties. The prepared new materials intended for application to the skin with high effectiveness. However, this paper has some problems that should be explained.

  1. Line 93, Section 2, Materials and Methods, containing 97.7% protein and 75% β-lactoglobulin in DM. The percentage of protein and lactoglobulin is greater than 100%? Please confirm.

Response: We thank the reviewer for pointing out the confusion. We have changed the first sentence of section 2.1 to:

"Whey protein isolate (WPI) (BiPRO, Davisco Foods Inter-national Inc., Eden Prairie, MN) containing 97.7% protein, of which 75% was β-lactoglobulin by dry mass (according to the manufacturer’s specification) was used."

  1. Line 110, 2.2. Preparation of Microparticles, stirring for 2h to ensure complete hydration of proteins. How to determine when complete hydration will occur in 2 hours?

Response: Sorry, it's our mistake. We have already removed this entry from the methodology.

  1. In 3.1. Characterization of Microparticles section. At the same ALG concentration, samples composed of 5% WPI seem to increase the water absorption rate. Why were only 4 and 5% chosen for the WPI ratio? Is there any reason?

Response: The selection of 4% and 5% WPI concentrations was deliberate, as these concentrations were found to facilitate the most effective formation of microparticles. Prior to final samples preparation, extensive testing was conducted across various concentration levels. It was observed that at higher concentrations, the polymer solutions exhibited increased viscosity, rendering them too thick to pass through the encapsulator nozzle effectively. Conversely, lower concentrations failed to induce microparticle formation or resulted in excessively delicate structures. Therefore, the concentrations of 4% and 5% were deemed optimal for use in the encapsulator, striking a balance between viscosity and flowability. Furthermore, the introduction of sodium alginate into the formulation played a crucial role in shaping the microparticles into spherical forms, enhancing their overall stability and suitability for the intended application.

  1. The description of Lines 233-235 was different from table 1:

“The samples composed of 5% WPI and 0.5% ALG displayed the highest swelling degree (approximately 830%)” Does it refer to swelling degree or water absorption rate %?

Response:  It refers to water absorption rate and it has been corrected. 

“In turn, the lowest water absorption was displayed by M(WPI 5% + ALG 1%) samples (approximately 670%)” à M(WPI 4% + ALG 1%) = 669 ± 57%; M(WPI 5% + ALG 1%) = 723 ± 38%

Response: It’s corrected to M(WPI 4% + ALG 1%). 

  1. Line 264-265, “Summarizing the analyses of…samples consisting of 5% WPI and 0.5% calcium alginate were selected ...”, but the following studies used M(WPI 4% + ALG 0.5%). Is there anything wrong?

Table 2 annotation “…films with microparticles (WPI 5% + ALG 0.5%)”, but the table content was M(WPI 4% + ALG 0.5%).

Response:  Yes, indeed, there was a slight oversight in our study. After careful review, it has come to our attention that the samples selected for further testing consisted of 4% WPI solutions and 0.5% sodium alginate solutions. We sincerely appreciate you paying close attention to detail in reviewing our article and bringing this error to our notice. Your keen observation underscores the importance of scrupulous control in scientific research, and we are grateful for your contribution to maintaining the accuracy and integrity of our work.

  1. This manuscript does not test any skin applications, and the "potential skin application" in the title is not appropriate.

Response: We completely agree with the Reviewer. In the future, we plan to test the application of materials on the skin, but it is too hasty to indicate this application at this stage in the title of the manuscript. The title has been changed.

Comments on the Quality of English Language

Moderate editing of English language required

Response: The native language of one of our co-authors (Dr. Timothy E.L. Douglas) is English. He checked and corrected the manuscript linguistically.

Reviewer 2 Report

Comments and Suggestions for Authors

Dear Editor, the primary purpose of this research was to produce microparticles based on whey protein isolate (WPI) and calcium alginate (ALG) containing Calendula officinalis flower extract and incorporate them into films composed of gelatin, WPI, and glycerol.

In this study microencapsulation of Calendula officinalis flower extracts have been prepared and the release of phenolic compounds has been studied. Is any advantage of microencapsulation, compared with nanoecapsulation (Polymers 2020, 12(7), 1542; https://doi.org/10.3390/polym12071542)  or Complexation of Curcumin with cyclodextrins (please see Antioxidants 2022, 11(11), 2271; https://doi.org/10.3390/antiox11112271) of similar extracts of phenolic compounds? A comparison of release profile of protection should be done and discussed in the paper.

Mechanical Tests. It is not clear if tensile strength measurements have been done, and which ISO or ASTM has been used.

What is the chemical structure of pot marigold extract? It is not clear in Figure 3, what authors are measured. It is s ingle compounds or a mixture of phenolic compounds?

It seems to me that the paper is now finished. Do the authors have additional results concerning the antioxidant properties of the prepared films? It was mentioned that Pot marigold extract was selected because of its beneficial effects on humans, such as antioxidant, anti-inflammatory, antimicrobial, and anti-viral properties. However, all these are claims without any prove.

Are any results to prove that: The vital advantages of microparticles are the protection of active substances against the damaging effects of external factors?

Comments on the Quality of English Language

Moderate editing of English language required

Author Response

Thank you very much for your valuable feedback. You have provided valuable guidance on every part of the article. These opinions have greatly benefited our team. Your guidance has allowed us to re-examine the article. In terms of structure, writing ideas, logic, and other aspects, it has given us a new and profound understanding of the following work steps. Thank you again for your help to our team.

Rewiever 2

Dear Editor, the primary purpose of this research was to produce microparticles based on whey protein isolate (WPI) and calcium alginate (ALG) containing Calendula officinalis flower extract and incorporate them into films composed of gelatin, WPI, and glycerol.

In this study microencapsulation of Calendula officinalis flower extracts have been prepared and the release of phenolic compounds has been studied. Is any advantage of microencapsulation, compared with nanoecapsulation (Polymers 2020, 12(7), 1542; https://doi.org/10.3390/polym12071542)  or Complexation of Curcumin with cyclodextrins (please see Antioxidants 2022, 11(11), 2271; https://doi.org/10.3390/antiox11112271) of similar extracts of phenolic compounds? A comparison of release profile of protection should be done and discussed in the paper.

Response: Many materials that are completely safe can become potentially problematic when in nano form, because when the particles are reduced in size, they can begin to function differently both chemically and physically, and potentially also biologically. There is the potential risk that nanoparticles might penetrate the outer layers of the skin to reach unintended sites in the body and interact with biological entities close to the molecular level. Due to the fact that in the future we plan to test the application of prepared materials on the skin of probands, we decided that a safer form of encapsulation of polyphenols would be microparticles. The publications indicated by the Rewiever indeed contribute a lot to the topic of polyphenols enclosed in nanoparticles for cosmetic purposes. For this reason, one of them was added to the references. 

Mechanical Tests. It is not clear if tensile strength measurements have been done, and which ISO or ASTM has been used.

Response:  In section 2.5.1, we provided a comprehensive description of the tensile tests conducted on the samples utilizing a testing machine. The stretching procedure continued until the samples reached their breaking point. It's noteworthy that although the samples were paddle-shaped, their dimensions deviated from those specified in the ASTM standard. Given our specialization in preparing samples composed of polymer blends with active substances, we devised a new approach for conducting tensile tests. This method was custom-tailored to suit our specific research focus and the unique characteristics of our sample compositions, thereby enhancing the reliability and relevance of our findings.

What is the chemical structure of pot marigold extract? It is not clear in Figure 3, what authors are measured. It is single compounds or a mixture of phenolic compounds?

Response: Calendula officinalis flower extract has a complex composition, including i.e. flavonoids, carotenoids, essential oil, and terpenoids. The release of extract entrapped in microparticles was investigated by evaluating the total phenolic content using a spectrophotometric method using the Folin-Ciocalteu test. The experimental calibration curve for the standard solution (gallic acid), prepared previously for loading capacity, was used. A new reference has been added to the description of the methodology. [V. L. Singleton, Joseph A. Rossi, Colorimetry of Total Phenolics with Phosphomolybdic-Phosphotungstic Acid Reagents, Am. J. Enol. Vitic. 16 (1965) 144-58].

It seems to me that the paper is now finished. Do the authors have additional results concerning the antioxidant properties of the prepared films? It was mentioned that Pot marigold extract was selected because of its beneficial effects on humans, such as antioxidant, anti-inflammatory, antimicrobial, and anti-viral properties. However, all these are claims without any prove.

Response: Response: In our research, we used purchased extract; we did not obtain it ourselves (Provital SA). Because this herb has been known for centuries and has been thoroughly tested, and we bought the extract as a ready-made substrate, we based the properties quoted by the Reviewer on literature data. Calendula officinalis flower extract, owing to its complex composition, possesses multiple activities, including antioxidant, antibacterial, antifungal, antiviral, anti-inflammatory, antioedematous, and wound healing activities, as well as immunostimulant, anticancer, hepatoprotective, and insecticidal properties. For these reasons, pot marigold is used in the treatment of numerous skin injuries, such as burns, ulcers, skin inflammations, eczema, bruises, cuts, and abrasions of the epidermis, rashes, skin wounds, frostbites, and other conditions. It is also used for cosmetic purposes for dry, flaky, cracked, and prone to infection or redness skin. Calendula officinalis is, therefore, a versatile and valuable herbal medicine used both externally and internally, which results from its rich and diverse chemical composition.

  • Khalid, K.; Teixeira da Silva, J. Biology of Calendula officinalis Linn.: Focus on pharmacology, biological activities and agronomic practices. Med. Aromat. Plant Sci. Biotechnol. 2012, 6, 12–27.
  • John, R.; Jan, N. Calendula Officinalis-An Important Medicinal Plant with Potential Biological Properties. Proc. Indian Natl. Sci. Acad. 2017, 93, 769–787.
  • Preethi, K.C.; Kuttan, G.; Kuttan, R. Antioxidant potential of an extract of Calendula officinalis flowers in vitro and in vivo. Pharm. Biol. 2006, 44, 691–697.]
  • Ukiya, M.; Akihisa, T.; Yasukawa, K.; Tokuda, H.; Suzuki, T.; Kimura, Y. Anti-inflammatory, anti-tumor-promoting, and cytotoxic activities of constituents of marigold (Calendula officinalis) flowers. J. Nat. Prod. 2006, 69, 1692–1696.
  • Efstratiou, E.; Hussain, A.I.; Nigam, P.S.; Moore, J.E.; Ayub, M.A.; Rao, J.R. Antimicrobial activity of Calendula officinalis petal extracts against fungi, as well as Gram-negative and Gram-positive clinical pathogens. Complement. Ther. Clin. Pract. 2012, 18, 173–176.
  • Jiménez-Medina, E.; Garcia-Lora, A.; Paco, L.; Algarra, I.; Collado, A.; Garrido, F. A new extract of the plant calendula officinalis produces a dual in vitro effect: Cytotoxic anti-tumor activity and lymphocyte activation. BMC Cancer 2006, 6, 1–14.
  • Fonseca, Y.M.; Catini, C.D.; Vicentini, F.T.M.C.; Nomizo, A.; Gerlach, R.F.; Fonseca, M.J.V. Protective effect of Calendula officinalis extract against UVB-induced oxidative stress in skin: Evaluation of reduced glutathione levels and matrix metalloproteinase secretion. J. Ethnopharmacol. 2010, 127, 596–601.
  • Chandran, P.K.; Kuttan, R. Effect of Calendula officinalis flower extract on acute phase proteins, antioxidant defense mechanism and granuloma formation during thermal burns. J. Clin. Biochem. Nutr. 2008, 43, 58–64.
  • Vargas, E.A.T.; Do Vale Baracho, N.C.; De Brito, J.; De Queiroz, A.A.A. Hyperbranched polyglycerol electrospun nanofibers for wound dressing applications. Acta Biomater. 2010, 6, 1069–1078.
  • Okuma, C.H.; Andrade, T.A.M.; Caetano, G.F.; Finci, L.I.; Maciel, N.R.; Topan, J.F.; Cefali, L.C.; Polizello, A.C.M.; Carlo, T.; Rogerio, A.P.; et al. Development of lamellar gel phase emulsion containing marigold oil (Calendula officinalis) as a potential modern wound dressing. Eur. J. Pharm. Sci. 2015, 71, 62–72.
  • Pedram Rad, Z.; Mokhtari, J.; Abbasi, M. Calendula officinalis extract/PCL/Zein/Gum arabic nanofibrous bio-composite scaffolds via suspension, two-nozzle and multilayer electrospinning for skin tissue engineering. Int. J. Biol. Macromol. 2019, 135, 530–543.
  • Aro, A.A.; Perez, M.O.; Vieira, C.P.; Esquisatto, M.A.M.; Rodrigues, R.A.F.; Gomes, L.; Pimentel, E.R. Effect of Calendula officinalis cream on achilles tendon healing. Anat. Rec. 2015, 298, 428–435
  • Akhtar, N.; Khan, B.A.; Haji, M.; Khan, S.; Ahmad, M.; Rasool, F.; Mahmood, T.; Rasul, A. Evaluation of various functional skin parameters using a topical cream of calendula officinalis extract. Afr. J. Pharm. Pharmacol. 2011, 5, 199–206.
  • Bernatoniene, J.; Masteikova, R.; Davalgiene, J.; Peciura, R.; Gauryliene, R.; Bernatoniene, R.; Majiene, D.; Lazauskas, R.; Civinskiene, G.; Velziene, S.; et al. Topical application of Calendula officinalis (L.): Formulation and evaluation of hydrophilic cream with antioxidant activity. J. Med. Plants Res. 2011, 5, 868–877.
  • Varka, E.-M.; Tsatsaroni, E.; Xristoforidou, N.; Darda, A.-M. Stability Study of O/W Cosmetic Emulsions Using Rosmarinus officinalis and Calendula officinalis Extracts. Open J. Appl. Sci. 2012, 2, 139–145.

Are any results to prove that: The vital advantages of microparticles are the protection of active substances against the damaging effects of external factors?

Response: Thank you for your valid comment. This sentence has been modified.

Comments on the Quality of English Language

Moderate editing of English language required

Response: The native language of one of our co-authors (Dr. Timothy E.L. Douglas) is English. He checked and corrected the manuscript linguistically.

Round 2

Reviewer 1 Report

Comments and Suggestions for Authors

Comments and Suggestions for Authors:

In this revised paper entitled “Whey protein isolate as a substrate to design Calendula officinalis flower extract controlled-release materials” (Article No. 2941990), most of the comments have been answered. Minor revisions could further enhance the manuscript, making it suitable for acceptance:

1.     The superscripts "a,b,c" in Table 1 should be annotated to indicate their respective meanings.

2.     Please ensure that the values in Table 2 are consistent with the text. For example, if Table 2 states "tensile strength 29 ± 2" the corresponding text should also mention "29 ± 2," rather than "29.1 ± 2.3 N."

3.     The botanical names in the literature should be italicized, as in Ref 23. The capitalization format of the titles of references should be consistent; for example, the formatting of ref 18 and 19 is inconsistent.

Comments on the Quality of English Language

Minor editing of English language required

Author Response

Dear Editor, Dear Reviewer,

Thank you for your very careful review of our paper and for the comments, corrections, and suggestions that ensued. A minor revision of the paper has been carried out to consider them all. In the process, we believe the paper has been significantly improved. Kindly excuse the delayed submission; we deeply regret the delay and inconvenience caused. We have strictly revised the manuscript according to the Reviewer's suggestions. 

Reviewer 2 Report

Comments and Suggestions for Authors

Dear editor,

In the revised manuscript the authors have well revised the paper and well replied the reviewers comments and questions. Now it is fine to be published.

Comments on the Quality of English Language

Minor editing of English language required

Author Response

Dear Editor, Dear Reviewer,

Thank you for your very careful review of our paper and for the comments, corrections, and suggestions that ensued.